# Functional Innovation through Gene Duplication Followed by Frameshift Mutation

**DOI:** 10.3390/genes13020190

**Published:** 2022-01-21

**Authors:** Baocheng Guo, Ming Zou, Takahiro Sakamoto, Hideki Innan

**Affiliations:** 1Key Laboratory of Zoological Systematics and Evolution, Institute of Zoology, Chinese Academy of Sciences, Beijing 100101, China; zoumingr@163.com; 2University of Chinese Academy of Sciences, Beijing 100049, China; 3Center for Excellence in Animal Evolution and Genetics, Chinese Academy of Sciences, Kunming 650223, China; 4Department of Evolutionary Studies of Biosystems, Graduate University for Advanced Studies, Hayama 240-0193, Kanagawa, Japan; sakamoto_takahiro@soken.ac.jp

**Keywords:** Ohno, gene duplication, frameshift mutation, *NOTCH2NL*, *ARHGAP11B*

## Abstract

In his influential book “*Evolution by Gene Duplication*”, Ohno postulated that frameshift mutation could lead to a new function after duplication, but frameshift mutation is generally thought to be deleterious, and thus drew little attention in functional innovation in duplicate evolution. To this end, we here report an exhaustive survey of the genomes of human, mouse, zebrafish, and fruit fly. We identified 80 duplicate genes that involved frameshift mutations after duplication. The frameshift mutation preferentially located close to the C-terminus in most cases (55/88), which indicated that a frameshift mutation that changed the reading frame in a small part at the end of a duplicate may likely have contributed to adaptive evolution (e.g., human genes *NOTCH2NL* and *ARHGAP11B*) otherwise too deleterious to survive. A few cases (11/80) involved multiple frameshift mutations, exhibiting various patterns of modifications of the reading frame. Functionality of duplicate genes involving frameshift mutations was confirmed by sequence characteristics and expression profile, suggesting a potential role of frameshift mutation in creating functional novelty. We thus showed that genomes have non-negligible numbers of genes that have experienced frameshift mutations following gene duplication. Our results demonstrated the potential importance of frameshift mutations in molecular evolution, as Ohno verbally argued 50 years ago.

## 1. Introduction

The significance of gene duplication for genetic innovation was first recognized by Stephens in the 1950s [1], and then popularized by Ohno in the 1970s [2]. Ohno postulated that one duplicate could evolve a new function by mutations, the scenario later well recognized as neofunctionalization. The evolution of gene duplication on a genomic scale has been extensively studied in a wide range of species, most of which argued the relative contribution of neofunctionalization and its derivatives, including subfunctionalization, by looking at amino acid evolution by point mutations and/or expression changes [3]. 

It is interesting to note that, in his seminal book *Evolution by Gene Duplication* [2], Ohno pointed out a potential role of frameshift mutations in neofunctionalization:

“Starting with a frame-shift mutation, a duplicate might acquire a new function, which is totally different from that assigned to the original gene. Admittedly, this is a one in a million chance, but in evolution, events with the odds of one in one million occurred time and time again”.

We re-recognized the importance of frameshift mutations from recent reports on duplication of the human *NOTCH* genes [4,5]. After the split with chimpanzee, a copy of human *NOTCH2* genes acquired a new function by a 4 bp deletion that introduced a fragment of completely new amino acid sequences. This duplicate, named *NOTCH2NL,* was further duplicated, and was speculated to have contributed to the rapid evolution of the human brain size. This striking case is exactly what Ohno pointed out, thereby motivating us to explore the contribution of frameshift mutations in the evolution of duplicate genes. Our literature survey found the human *ARHGAP11* genes as another convincing case [6,7,8,9,10]. *ARHGAP11B* arose from duplication of *ARHGAP11A* in the human lineage after separation from the chimpanzee lineage, and evolved the new function of increasing basal progenitor amplification during neocortex development. The new function of *ARHGAP11B* is acquired by a 55 deletion in its mRNA that leads to loss of RhoGAP activity by GAP domain truncation, and addition of a human-specific carboxy-terminal amino acid sequence [6,7].

Beside these clear cases, there have been very few documentations of frameshift mutations involved in the adaptive evolution. Several genomic surveys for frameshifts between duplicate genes have been reported [11,12,13,14] in animals. However, most of the identified ones were involved in complex alternative splicing and exon shuffling, and none was a convincing case of Ohno’s prediction, except for one case in the rodent lineage (the functional divergence between *NKG2A* and *NKG2C*) [11]. In plants, putative frameshift mutations were found in the MADS-box gene family, the B-function *DEF/AP3* subfamily, the A-function *SQUA/AP1* subfamily, and the E-function *AGL2* subfamily, all of which are involved in the specification of organ identity during flower development [15]. 

We here report an exhaustive survey of the genomes of human (*Homo sapiens*), mouse (*Mus musculus*), zebrafish (*Danio rerio*), and fruit fly (*Drosophila melanogaster*) to identify frameshift mutation in duplicates. We evaluate and argue for the contribution of the scenario of duplication–frameshift–neofunctionalization in these four diverse organisms and its evolutionary consequences. 

## 2. Materials and Methods

Genomic data was retrieved from Ensembl (release 88). To identify putative duplicates, TBLASTX [16] of all coding sequences (CDS) against themselves was first performed with an E-value cutoff of 1 × 10^−7^ in each genome. This meant that all transcripts for each gene were included to exclude frameshifts that were introduced by alternative splicing. Then, high-scoring pairs (HSP) with both overlap and identification ≥ 70% were selected as putative duplicates using custom Perl scripts. Next, a protein sequence from one CDS of a HSP was mapped to another CDS of the HSP with GeneWise [17] for each HSP to identify HSPs with frameshift mutation. HSPs with frameshift mutations were selected as potential candidates. Considering that alternative splicing commonly results in frameshift mutation among different transcripts in a gene, only candidate HSPs in each of which all transcripts from one gene against all transcripts from another gene with consistently the same frameshift mutation were kept. As such, redundancy of the candidate HSPs with frameshifts was also reduced from transcript to gene level. Thirdly, duplication events of the identified duplicated pair with frameshift mutations were confirmed with a comparative genomics profile in Ensembl. Finally, the candidate duplicated pairs with frameshift mutations were manually checked with great care for orthology, synteny, and sequence alignment by utilizing multiple sequence alignments of orthologs and paralogs in Ensembl. 

To best investigate the evolutionary trajectory of frameshift mutations in duplicates, frameshift mutations were defined by considering the outgroup; that is, the sequence from the most closely related species free from the duplication event. In cases without available outgroup sequences, frameshift mutations were defined in the shorter copy by considering characteristics of the copy with frameshift mutations in cases with outgroup sequences. First, location and length of frameshift mutations were sketched. An example is shown in Figure 1. The region in which both copies used the same reading frame was referred to as a common frame region. The region using the new reading frame in the derived copy was called the frameshifted region, and its corresponding part in the original copy was called the original frame region. Then, the divergence and selection pressure between duplicates in their common frame and frameshift region were characterized, respectively, by calculating the numbers of synonymous (*K*_s_) and nonsynonymous substitutions (*K*_a_) per site using codeml with the pairwise model in PAML4 [18]. Specifically, to estimate the selection pressure, an ancestry sequence was first reconstructed based on duplicates and their closely related outgroup sequence at the nucleotide level. Then, *K*_a_ and *K*_s_ in each copy of duplicates were calculated by comparison between the duplicate sequence and its ancestry sequence for the common frame region and the frameshifted region, respectively. For the calculation of *K*_a_ and *K*_s_, we used the original reading frame.

The gene expression information was retrieved from GTEx (https://www.gtexportal.org/home/, accessed on 2 April 2020) for human, and from MGI (http://www.informatics.jax.org/, accessed on 2 April 2020) for mouse. The gene expression information for zebrafish was obtained by mapping RNA-seq data from SRA to its genome with HISAT and StringTie pipeline [19]. The statistics were done in R version 3.3.3.

## 3. Results and Discussion

We identified in total 80 pairs of duplicate genes with frameshift mutations between them in these species (see Materials and Methods): 17 in human (*NOTCH2* genes were not found because they were not annotated in Ensembl release 88), 30 in mouse, 31 in zebrafish, and 2 in fruit fly (Table 1). For each pair of duplicates, we searched for an outgroup sequence, which allowed us to determine which copy underwent frameshift mutation(s) (referred to as the derived copy) and which was the original copy. To do so, we computed *K_s_C_* and *K_a_C_*, the numbers of synonymous and nonsynonymous substitutions per site in common frame regions, respectively. We predicted that *K_s_C_* roughly gave the age of the duplication, while knowing the gene conversion between them retarded the divergence [20]. We then searched for an outgroup sequence such that *K_s_C_* was sufficiently smaller than the typical species divergence at synonymous sites. We further looked at the synteny to confirm their orthology. We determined the original and derived copies for 53 cases (Appendix A).

We first looked at the number and the locations of frameshift mutations. We found that most cases (69/80) experienced only one frameshift mutation, while the others underwent multiple frameshift mutations (11/80). The 69 cases with one frameshift mutation were categorized into type-C and -N classes according to whether the frameshift mutation changed the reading frame in the C- or N-terminus of the amino acid sequence. Only a representative case is shown for type-C and -N classes in Figure 2A,B, respectively. We found that the majority was categorized into the type-C class (51 vs. 18, *p* = 0.0001, binomial test). The enrichment of type-C was reasonable, because it can be explained by a single mutation alone; that is, the reading frame downstream of the mutation was changed by the mutation. In contrast, although only one mutation was detected for a type-N case, it should be noted that another mutation upstream of the start codon is needed to create a type-N (otherwise, it becomes a type-C). 

If we used those with an outgroup sequence available (34 type-C and 12 type-N cases), we found that the frameshifted region was shorter than the original frame region in >80% cases (27/34 and 11/12 cases for type-C and -N, respectively). If the entire coding region of the original copy was assigned to the unit interval (0,1), the average relative length of the frameshifted region (0.16 and 0.10 for type-C and -N cases, respectively) was shorter than the original frame region (0.30 and 0.21 for type-C and -N cases, respectively), and the difference was significant (*p* = 0.05 and 0.01 for type-C and -N cases, respectively; permutation test). It was suggested that a frameshift mutation likely resulted in a shorter coding region, which was also seen in *NOTCH2NL* [4,5] and *ARHGAP11B* [6,7]. 

Figure 3A shows the distribution of relative locations of frameshift mutations, where we used these 46 cases with an outgroup available and also 23 cases with no outgroup (in total, 51 type-C and 18 type-N cases). For the latter cases, given the above results, we assumed that the shorter ones were derived copies, with one exceptional case M22 (Appendix A). It was found that their distributions were not uniform, and that most frameshift mutations were found close to the stop codon of the original copy for type-C cases and close to the start codon for type-N cases. Figure 3B shows the distribution of relative length of the frameshifted region, indicating that the length of the frameshifted region in general was shorter than 20% of the entire region.

The patterns of the 11 cases with multiple frameshift mutations greatly varied; for 7 of these, we obtained an outgroup sequence, and the alignment of the original and derived copies are shown in Figure 2C. The 11 cases included 3 Type-C cases with 2 mutations (Z01, Z18, and Z20, of which Z20 is shown in Figure 2C). These cases were most likely created as follows. A first frameshift mutation created a frameshifted region in the C-terminus, and a second frameshift mutation occurred downstream of the first one, causing another new reading frame. A similar pattern was observed in six cases (M13, M18*, M25, M27*, Z10, and F01*, of which those with a star are shown in Figure 2C), where more than one mutation created a frameshifted region in the middle of the coding region (type-M). For these cases, it is reasonable to consider that a first mutation created a frameshifted region in the C-terminus, and a secondary frameshift mutation occurred downstream of the first one. The difference was that a secondary mutation reversed the reading frame back to the original reading frame. Other complex cases included H03 (Figure 2C), which experienced two frameshift mutations, resulting in two distinct frameshifted regions, one in the C-terminus and the other in the N-terminus. 

Figure 3C,D show the distributions of *K_s_C_* and *K_a_C_*/*K_s_C_* for 51 type-C, 18 type-N, and 11 multiple-mutation cases. We found that most cases had *K_s_C_* < 0.3, while some type-C cases had *K_s_C_* > 0.3. The distributions of *K_a_C_*/*K_s_C_* for the three types seemed similar: the majority had *K_a_C_*/*K_s_C_* < 1, with an average of 0.65. The exact same distributions are shown with labels of the four species (Figure 3E,F). In Appendix A, the configurations of common frame and frameshifted regions for all 80 cases are shown. 

Provided that the most likely fate of a duplicate is to become a pseudogene, a major question is whether those frameshift mutations played a meaningful role in functional genes, or they were merely random deleterious mutations being accumulated on the way to becoming a pseudogene. To address this question, we first asked if those frameshifted duplicates were functional in the current genome. We looked over the list of the duplicates, and found that most of them were well-annotated genes (78/80; Table 1), including *ARHGAP11* (our ID: H08) in human, as mentioned above [6,7]; oocyte-specific homeobox (*Obox*) genes (our ID: M16) [21] and vomeronasal type-1 receptor (*Vmn1r*) genes (our ID: M21, 22, and 24) [22] in mouse; and alpha-actinin genes (our ID: Z30) in zebrafish [23]. We also confirmed the presence of expression in 78/80 cases, and all of them showed divergence in the expression amount and/or pattern, further supporting their functional importance (Table 1).

Further evidence for stable preservation of the derived frameshifted copy should be seen in their DNA sequences. First, we found that 37/80 cases had *K_s_C_* > 0.1. A relatively large divergence between paralogs (i.e., *K_s_C_* > 0.1) should indicate that these derived copies had been preserved as functional genes for a significant amount of time, while maintaining a low *K_a_C_*. This meant that they had successfully escaped pseudogenization, because pseudogenization usually occurs in a relatively short time after duplication [3]. 

Next, if frameshift mutations significantly contributed to the acquisition of novel functions, we predicted we would observe accelerated amino acid substitutions in the derived copy, particularly in the frameshifted region. A testable prediction would be that the *K_a_*/*K_s_* ratio was exceeded on the lineage, leading to the derived copy (denoted by *K_a_D_*/*K_s_D_*), particularly in the frameshifted region (denoted by *K_a_DF_*/*K_s_DF_*). We tested this for 49 cases with a reliable outgroup sequence available (Appendix A), for which we estimated the ancestral sequence for each pair of duplicates, and synonymous and nonsynonymous divergences from the ancestral sequence to the original and derived copies were computed (denoted by *K_a_O_* and *K_s_O_* for the original copy and *K_a_D_* and *K_s_D_* for the derived copy). Note that we used the original reading frame for this calculation. We found one case (Z13) that exhibited *K_a_D_*/*K_s_D_* significantly larger than 1 for the entire region (*K_a_D_* = 0.058 vs. *K_s_D_* =0.000, *p* < 0.03). By focusing on the frameshifted region, we tested the null hypothesis of an equal *K_a_*/*K_s_* ratio for the original and derived lineages (i.e., *K_a_OF_*/*K_s_OF_* = *K_a_DF_*/*K_s_DF_*), and found one case (Z13) with a significant deviation (*K_a_OF_*/*K_s_OF_* = 0.000, *K_a_DF_*/*K_s_DF_* = 2.423, *p* < 0.0001). One might suspect that the observation might be explained by relaxed selection of the derived copy, but this should not be the case, due to a low *K_a_*/*K_s_* ratio in the common frame region (*K_a_*/*K_s_* = 0.27), indicating that the *K_a_*/*K_s_* ratio was elevated only in the frameshifted region, and selective constraint remained in the common frame region. We were unable to detect statistically significant results for the other cases, mostly due to a small number of substitutions because the frameshifted regions were generally very short (Appendix A). If we pooled all cases, we obtained *K_a_OF_*/*K_s_OF_* = 0.374, *K_a_DF_*/*K_s_DF_* = 1.087, suggesting that the amino acid substitution rate was on average large on the lineage, leading to the derived copy in the frameshifted region (but not in the common frame region: *K_a_OC_*/*K_s_OC_* = 0.449, *K_a_DC_*/*K_s_DC_* = 0.519).

Thus, for the majority of the cases, there were reasons that supported significant functional roles of the derived copies in the current genome. We then argued the implications of these results on the potential roles of frameshift mutations. First, it is interesting to point out that if a frameshift mutation occurred in an intact coding gene, it always resulted in type-C; that is, the reading frame downstream of the mutation was modified. All other types (i.e., types-N, -M, etc.) needed subsequent mutations. This should be the reason why the most frequent type was type-C. In type-C cases, frameshift mutations for the C-type were enriched near the C-terminus. Assuming mutation occurred randomly along a gene, the skewed observed distribution should indicate that frameshift mutations were in general deleterious and selected against, especially when enough occurred to change the majority of the reading frame. The condition that a frameshift mutation directly contributes to adaptive evolution is that it confers a selective advantage by creating a new reading frame. Most likely, it should occur without changing the major part of the coding gene, otherwise it would be too deleterious to survive. If the new reading frame is beneficial, the frameshift mutation may be favored by selection and preserved in the genome. Another possible scenario would be that the new reading frame is nearly neutral, so that it is not immediately selected out. Then, subsequent amino acid changing mutations could occur in the frameshifted region to create a beneficial amino acid sequence. Such a secondary mutation has to occur before the first mutation is selected out. 

In a similar vein, other kinds of secondary mutations could confer selective advantages, and their outcomes include type-N and -M. Suppose a first frameshift mutation creates a type-C gene. When a secondary frameshift mutation occurs downstream of the first one, it will become a type-M gene if it changes the downstream reading frame to the original frame (i.e., a frameshifted region in the middle of the coding region is observed). As mentioned earlier, to explain a type-N gene, a secondary mutation is needed to create a new translation start site upstream of the first mutation, by which a type-C gene immediately turns out to be a type-N gene. Provided that we observed all frameshift mutations close to the N-terminus (Figure 2A), one possible scenario was that these first frameshift mutations killed the original function and produced pseudogenes temporarily, and secondary rescue mutations recovered the original reading frame, perhaps quickly, resulting in type-N genes. In this work, we found some cases (2/80) with multiple frameshift mutations that could be explained by this scenario. Considering that the rate of such rescue mutations was very low, our observation might be difficult to explain by neutral evolution alone. In addition, it should be considered that the results of our study critically depended on genomic assembly and annotation quality, and sequencing and gene prediction errors may affect studies such as ours here. We expect that long-read sequencing could be utilized to validate our findings in future studies.

## 4. Conclusions

We thus demonstrated that genomes have non-negligible numbers of genes that have experienced frameshift mutations following gene duplication. In most cases, the majority of the original amino acid sequence was kept, and a small part was modified. A frameshift mutation produced a random amino acid sequence downstream until a premature stop codon arises. It should be very rare that such a random sequence provides a novel function for neofunctionalization, but on a genomic scale, we observed a number of cases that benefited from frameshift mutations. Most especially, the studies of *NOTCH* genes [4,5] and *ARHGAP11* genes [6,7,8,9,10] in human suggest that the functional innovation through gene duplication followed by frameshift mutation could play an important role in adaptive evolution of functional traits. Our results demonstrated the potentially importance of frameshift mutations in molecular evolution, as Ohno verbally argued 50 years ago.

## Figures and Tables

**Figure 1 genes-13-00190-f001:**
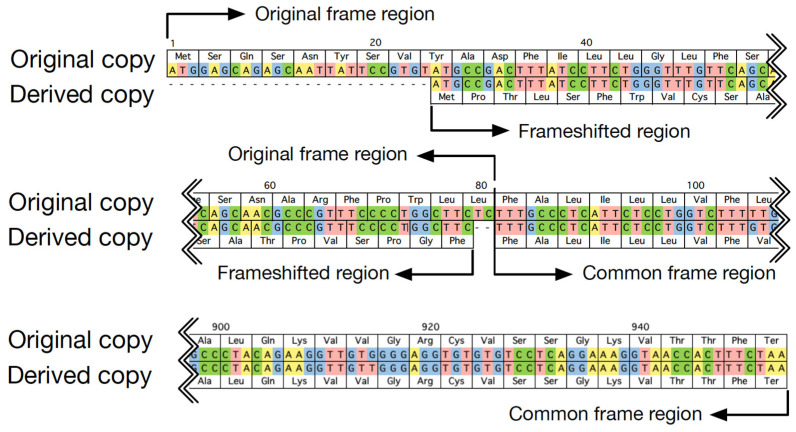
An example of duplicated genes, with one copy having experienced a frameshift mutation (H10, *OR2T7,* and *OR2T27* in human in Table 1).

**Figure 2 genes-13-00190-f002:**
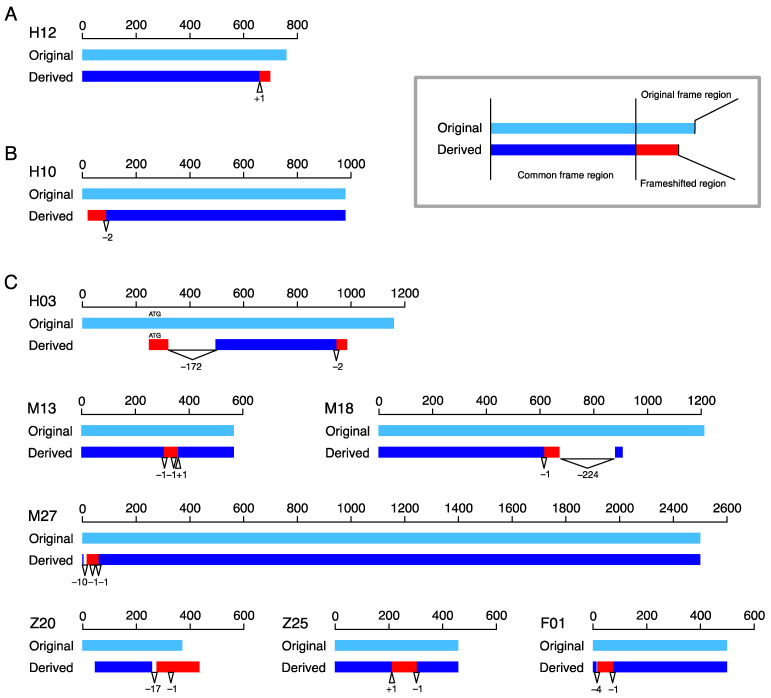
Locations of typical frameshift mutations along their coding sequences in bp from the start codon. The original and derived copies are presented in light blue and blue, with frameshifted regions in red. Triangles with positive numbers are insertions, and reverse triangles with negative numbers are deletions. The numbers are the sizes of the indels. (**A**) One representative case for type-C. (**B**) One representative case for type-N. (**C**) Seven cases involving multiple frameshift mutations: H12, *CASTOR3,* and *CASTOR2* in human; H10, *OR2T7,* and *OR2T27* in human; H3, *MICA,* and *MICA* in human; M13, *Rpl-ps1,* and *Rpl-ps6* in mouse; M18, *Lipo5,* and *Lipo3* in mouse; M27, *Pcdhb7,* and *Pcdhb9* in mouse; Z20 and Z25 in zebrafish (gene names are unavailable); and F01, CSNK2B, and CSNK2B in fruit fly.

**Figure 3 genes-13-00190-f003:**
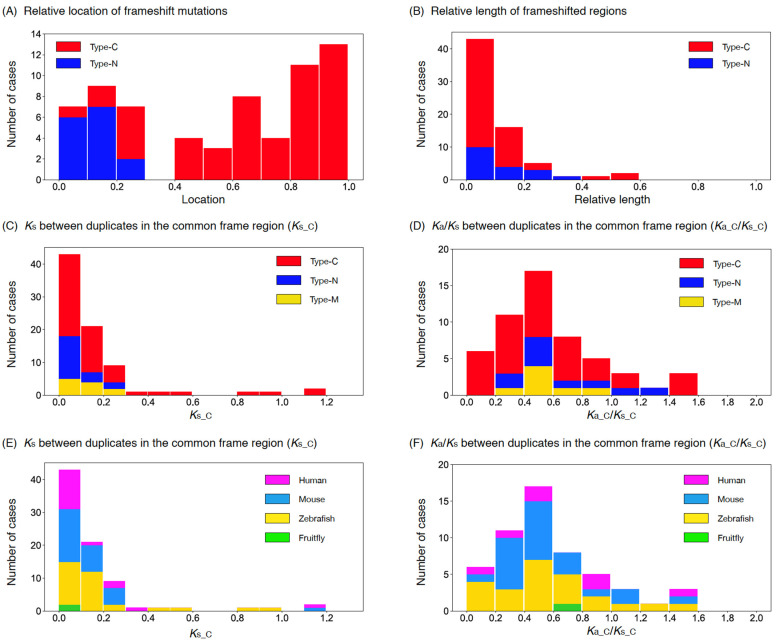
(**A**) Distribution of the locations of frameshift mutations mapped on the original copy rescaled to the (0,1) interval. (**B**) Distribution of the size of frameshifted region relative to the original copy. (**C**) Distribution of *K_s_C_* for type-C, -N, and -M. (**D**) Distribution of *K_a_C_*/*K_s_C_* for type-C, -N, and -M. Data for the four species are pooled. (**E**) Distribution of *K_s_C_* for human, mouse, zebrafish, and fruit fly. (**F**) Distribution of *K_a_C_*/*K_s_C_* for human, mouse, zebrafish, and fruit fly. Data for type-C, -N, and -M are pooled. The data were binned with a window of size 0.1 in all panels.

**Table 1 genes-13-00190-t001:** Duplicated genes with frameshift mutations.

Species	Group ID	Gene ID of Original Copy	Gene ID of Derived Copy	Gene Name of Original Copy	Gene Name of Derived Copy	Gene ID of Outgroup	No. of Frameshift Mutations	Type	Length (No. of Amino Acids)	Expression Divergence
Human	H01	ENSG00000235233.8	ENSG00000204520.12	*MICA*	*MICA*	ENSPTRT00000033163.5	1	C ****	15/223/385 ***	Quantitative level
	H02	ENSG00000235233.8	ENSG00000231225.9	*MICA*	*MICA*	ENSPTRT00000033163.5	1	C	15/289/385	Quantitative level
	H03	ENSG00000235233.8	ENSG00000233051.9	*MICA*	*MICA*	ENSPTRT00000033163.5	2	N&C	27/194/385&15/194/385	Quantitative level
	H04	ENSG00000233439.7	ENSG00000206458.9	*PSORS1C1*	*PSORS1C1*	ENSPTRT00000066839.2	1	C	24/63/152	Quantitative level
	H05	ENSG00000170122.5	ENSG00000184492.6	*FOXD4*	*FOXD4L1*	ENSMICG00000036465	1	C	116/408/439	Quantitative level
	H06	ENSG00000153779.10	ENSG00000176679.8	*TGIF2LX*	*TGIF2LY*	ENSPCOT00000008535.1	1	C	37/185/241	Pattern/Quantitative level *****
	H07	ENSG00000170122.5	ENSG00000273514.1	*FOXD4*	*FOXD4L6*	ENSMICG00000036465	1	C	55/417/439	Quantitative level
	H08	ENSG00000275568.4	ENSG00000187951.10	*ARHGAP11A*	*ARHGAP11B*	ENSPTRT00000012671.3	1	C	47/267/1023	Pattern/Quantitative level
	H09	ENSG00000204149.10	ENSG00000204172.12	*AGAP6*	*AGAP9*	ENSPTRG00000029891	1	C	15/658/686	Quantitative level
	H10	ENSG00000187701.3	ENSG00000281395.1	*OR2T27*	*OR2T7*	ENSGGOG00000003840	1	N	17/308/317	NA
	H11	ENSG00000211678.2	ENSG00000211676.2	*IGLJ3*	*IGLJ2*	NA	1	N	16/47/50	Pattern/Quantitative level
	H12	ENSG00000274070.1	ENSG00000239521.7	*CASTOR2*	*CASTOR3*	NA	1	N	36/163/329	Quantitative level
	H13	ENSG00000258405.9	ENSG00000221874.4	*ZNF578*	*ZNF816-ZNF321P*	ENSCCAG00000033386	1	C	12/233/253	Quantitative level
	H14	ENSG00000134545.13	ENSG00000183542.5	*KLRC4*	*KLRC4*	ENSMUST00000053708.8 *	1	C	15/158/233	Pattern/Quantitative level
	H15	ENSG00000134545.13	ENSG00000255819.7	*KLRC1*	*KLRC4-KLRK1*	ENSMUST00000053708.8	1	C	15/150/228	Quantitative level
	H16	ENSG00000182816.8	ENSG00000186980.6	*KRTAP13-2*	*KRTAP23-1*	NA	1	C	26/65/175	Pattern/Quantitative level
	H17	ENSG00000152086.8	ENSG00000243910.7	*TUBA3E*	*TUBA4B*	NA	1	C	40/241/450	Quantitative level
Mouse	M01	ENSMUSG00000060816.2	ENSMUSG00000062546.4	*Vmn1r52*	*V1ra8*	ENSRNOT00000086882.1	1	C	13/279/309	Pattern/Quantitative level
	M02	ENSMUSG00000094918.3	ENSMUSG00000096641.6	*Gm8765*	*Gm904*	ENSRNOT00000060670.1	1	C	16/102/1017	Pattern/Quantitative level
	M03	ENSMUSG00000066487.3	ENSMUSG00000090544.2	*Gm5786*	*Gm6576*	ENSRNOT00000019508.6	1	N	13/274/302	Pattern/Quantitative level
	M04	ENSMUSG00000066487.3	ENSMUSG00000044533.15	*Gm5786*	*Rps2*	ENSRNOT00000019508.6	1	C	26/293/302	Pattern/Quantitative level
	M05	ENSMUSG00000091733.1	ENSMUSG00000096372.1	*Gm8094*	*Gm8138*	ENSRNOT00000046644.4 *	1	N	51/213/211	Pattern/Quantitative level
	M06	ENSMUSG00000072066.6	ENSMUSG00000055228.7	*6720489N17Rik*	*Zfp935*	ENSRNOT00000061526.2	1	C	62/106/353	Pattern/Quantitative level
	M07	ENSMUSG00000099974.1	ENSMUSG00000053820.4	*Bcl2a1d*	*Bcl2a1c*	ENSRNOT00000039850.3	1	C	26/128/172	Pattern/Quantitative level
	M08	ENSMUSG00000109516.1	ENSMUSG00000109396.1	*Gm6882*	*Gm4565*	ENSRNOT00000058760.2	1	C	13/301/313	NA
	M09	ENSMUSG00000091477.1	ENSMUSG00000072595.2	*Gm5799*	*4930503E14Rik*	NA	1	C	15/177/201	Pattern/Quantitative level
	M10	ENSMUSG00000096446.1	ENSMUSG00000079244.3	*Gm8104*	*Gm5622*	ENSRNOT00000042743.5	1	C	19/167/198	Pattern/Quantitative level
	M11	ENSMUSG00000099115.1	ENSMUSG00000092086.1	*Gm17190*	*Gm6793*	NA	1	N	28/357/351	Pattern/Quantitative level
	M12	ENSMUSG00000031320.9	ENSMUSG00000098559.1	*Rps4x*	*Gm15013*	ENSRNOT00000076978.3	1	C	11/263/266	Pattern/Quantitative level
	M13	ENSMUSG00000062456.3	ENSMUSG00000081906.2	*Rpl9-ps6*	*Rpl9-ps1*	ENSRNOT00000052231.4	3	M	15/191/192	Pattern/Quantitative level
	M14	ENSMUSG00000047980.6	ENSMUSG00000091411.1	*9230110F15Rik*	*Gm5916*	ENSRNOT00000032208.3	1	C	18/102/117	Pattern/Quantitative level
	M15	ENSMUSG00000099294.1	ENSMUSG00000081607.2	*Gm11214*	*Gm15294*	NA	1	C	20/183/112	Pattern/Quantitative level
	M16	ENSMUSG00000055942.13	ENSMUSG00000074369.12	*Obox7*	*Obox2*	ENSRNOT00000045660.2	1	C	27/151/218	Pattern/Quantitative level
	M17	ENSMUSG00000094472.1	ENSMUSG00000094856.1	*Gm21897*	*Gm21962*	ENSRNOT00000091672.1	1	C	19/338/500	Pattern/Quantitative level
	M18	ENSMUSG00000024766.14	ENSMUSG00000086875.1	*Lipo3*	*Lipo5*	ENSRNOT00000035013.4	2	M	20/233/399	Pattern/Quantitative level
	M19	ENSMUSG00000108596.1	ENSMUSG00000062997.6	*Gm49368*	*Rpl35*	NA	1	C	48/125/123	Pattern/Quantitative level
	M20	ENSMUSG00000067919.8	ENSMUSG00000058186.13	*Rex2*	*Zfp980*	ENSRNOT00000081537.1	1	C	11/645/685	Pattern/Quantitative level
	M21	ENSMUSG00000061829.3	ENSMUSG00000071490.3	*Vmn1r214*	*Vmn1r212*	NA	1	C	55/357/367	Pattern/Quantitative level
	M22	ENSMUSG00000061829.3	ENSMUSG00000060024.1	*Vmn1r214* **	*Vmn1r213*	NA	1	N	79/384/367	Pattern/Quantitative level
	M23	ENSMUSG00000078495.10	ENSMUSG00000078496.9	*Zfp984*	*Zfp982*	ENSRNOT00000034746.6	1	C	16/368/547	Pattern/Quantitative level
	M24	ENSMUSG00000100296.1	ENSMUSG00000069289.1	*Vmn1r205*	*Vmn1r203*	MGP_PahariEiJ_T0040451.1	1	C	11/311/316	Pattern/Quantitative level
	M25	ENSMUSG00000095638.1	ENSMUSG00000093922.1	*Gm8888*	*Rpl36-ps4*	NA	3	N&M	34/117/126&9/117/126	Pattern/Quantitative level
	M26	ENSMUSG00000096515.5	ENSMUSG00000091008.1	*Igkv14-100*	*Gm17472*	ENSRNOT00000087773.1	1	C	14/117/117	Pattern/Quantitative level
	M27	ENSMUSG00000051242.1	ENSMUSG00000045062.4	*Pcdhb9*	*Pcdhb7*	ENSMOCT00000026210.1	3	M	20/829/828	Pattern/Quantitative level
	M28	ENSMUSG00000067199.4	ENSMUSG00000070526.2	*Frat1*	*Frat3*	ENSNGAT00000028350.1	1	C	17/283/274	Pattern/Quantitative level
	M29	ENSMUSG00000027925.2	ENSMUSG00000050635.1	*Sprr2j-ps*	*Sprr2f*	NA	1	C	56/76/109	Pattern/Quantitative level
	M30	ENSMUSG00000035783.8	ENSMUSG00000099104.1	*Acta2*	*Gm17087*	NA	1	C	21/156/221	Pattern/Quantitative level
Zebrafish	Z01	ENSDARG00000099789.1	ENSDARG00000095189.1	*si:ch211-76m11.7*	*si:ch211-127n13.2*	NA	2	C	20/157/173	Pattern/Quantitative level
	Z02	ENSDARG00000090975.3	ENSDARG00000103701.1	*si:dkey-62k3.5*	NA	XP_025757774.1	1	N	29/91/133	Pattern/Quantitative level
	Z03	ENSDARG00000092779.1	ENSDARG00000102941.1	*si:rp71-23d18.4*	*si:ch211-249c2.1*	NA	1	N	14/193/229	Pattern/Quantitative level
	Z04	ENSDARG00000097099.1	ENSDARG00000095593.1	*si:ch211-108d22.1*	*si:ch211-134c9.2*	XP_018956261.1	1	N	15/63/66	Pattern/Quantitative level
	Z05	ENSDARG00000102028.1	ENSDARG00000102853.1	*si:dkey-151g22.1*	*Si:rp71-7l19.2*	XP_018942195.1	1	N	21/233/266	Pattern/Quantitative level
	Z06	ENSDARG00000092625.2	ENSDARG00000092202.2	*HTRA2*	*HTRA2*	ENSIPUT00000009250.1	1	C	15/194/214	Pattern/Quantitative level
	Z07	ENSDARG00000095545.2	ENSDARG00000094878.2	*HTRA2*	*si:dkey-84o3.6*	ENSIPUT00000009250.1	1	C	34/198/203	Pattern/Quantitative level
	Z08	ENSDARG00000074279.1	ENSDARG00000078586.3	NA	NA	CI01000026_05016210_05017522 *	1	N	22/221/233	Pattern/Quantitative level
	Z09	ENSDARG00000095444.1	ENSDARG00000093963.2	*si:dkey-112g5.16*	*si:dkey-112g5.12*	ENSIPUT00000009250.1	1	N	24/202/268	Pattern/Quantitative level
	Z10	ENSDARG00000092512.1	ENSDARG00000093579.1	*si:dkey-93l1.6*	*si:dkey-80c24.4*	XP_018949576.1	2	M	34/218/462	Pattern/Quantitative level
	Z11	ENSDARG00000090975.3	ENSDARG00000099246.1	*si:dkey-62k3.5*	NA	XP_025757774.1	1	N	20/108/133	Pattern/Quantitative level
	Z12	ENSDARG00000095532.1	ENSDARG00000095026.1	*si:dkey-58f10.13*	*si:dkey-58f10.14*	NA	1	C	63/216/232	Pattern/Quantitative level
	Z13	ENSDARG00000092512.1	ENSDARG00000091890.1	*si:dkey-93l1.6*	*si:dkeyp-67e1.3*	XP_018949576.1	1	C	54/288/462	Pattern/Quantitative level
	Z14	ENSDARG00000099200.1	ENSDARG00000077910.4	*zgc:123103*	*si:zfos-1897c11.1*	ENSIPUT00000032654.1	1	N	12/257/347	Pattern/Quantitative level
	Z15	ENSDARG00000043445.5	ENSDARG00000093845.3	*si:ch211-152f2.3*	*si:ch211-1f22.12*	XP_016119820.1	1	C	37/360/384	Pattern/Quantitative level
	Z16	ENSDARG00000094001.3	ENSDARG00000093588.1	*si:dkey-6a5.3*	*si:dkey-234l24.8*	NA	1	C	15/70/90	Pattern/Quantitative level
	Z17	ENSDARG00000104631.1	ENSDARG00000078683.3	*RNF14*	*RNF14*	ENSIPUT00000017639.1	1	C	77/413/387	Pattern/Quantitative level
	Z18	ENSDARG00000094001.3	ENSDARG00000094329.1	*si:dkey-6a5.3*	*si:dkey-234l24.1*	NA	2	C	18/73/90	Pattern/Quantitative level
	Z19	ENSDARG00000017984.10	ENSDARG00000096625.1	*zgc:172106*	*si:ch211-286b5.8*	NA	1	C	13/332/394	Pattern/Quantitative level
	Z20	ENSDARG00000043894.5	ENSDARG00000102034.1	*si:ch211-214k5.3*	*si:ch211-214k5.3*	XP_016341397.1	2	C	38/124/124	Pattern/Quantitative level
	Z21	ENSDARG00000026704.6	ENSDARG00000097373.2	*ftr72*	*ftr90*	XP_016407654.1	1	N	18/489/550	Pattern/Quantitative level
	Z22	ENSDARG00000078195.2	ENSDARG00000104993.2	*fhit*	*si:ch211-63i20.3*	XP_018967896.1	1	C	11/110/150	Pattern/Quantitative level
	Z23	ENSDARG00000100941.1	ENSDARG00000086498.3	*clca1*	*si:ch211-116o3.3*	CI01000308_00041387_00048067	1	C	9/229/907	Pattern/Quantitative level
	Z24	ENSDARG00000102866.1	ENSDARG00000105247.1	*CR391991.1*	*CR391991.5*	NA	1	C	35/206/223	Pattern/Quantitative level
	Z25	ENSDARG00000095057.2	ENSDARG00000070858.4	*si:dkey-103d23.3*	*si:busm1-105l16.2*	XP_016375730.1	2	M	34/226/267	Pattern/Quantitative level
	Z26	ENSDARG00000097997.1	ENSDARG00000089747.3	*si:ch211-272b8.7*	*si:zfos-754c12.2*	NA	1	N	42/330/391	Pattern/Quantitative level
	Z27	ENSDARG00000068363.5	ENSDARG00000092930.1	*si:ch211-132b12.3*	*si:ch211-132b12.4*	CI01000028_05937526_05949610	1	N	10/112/577	Pattern/Quantitative level
	Z28	ENSDARG00000069869.3	ENSDARG00000103341.1	*zgc:113030*	*si:ch211-110e21.4*	NA	1	C	16/215/240	Pattern/Quantitative level
	Z29	ENSDARG00000099359.1	ENSDARG00000086495.2	*CR855320.1*	*BX546500.1*	KKF19921.1 *	1	C	12/1344/1430	Pattern/Quantitative level
	Z30	ENSDARG00000001431.10	ENSDARG00000104713.1	*actn3b*	*zgc:165653*	ENSEBUT00000022154.1	1	C	32/151/898	Pattern/Quantitative level
	Z31	ENSDARG00000101249.2	ENSDARG00000105342.1	*zgc:165555*	*si:dkey-23a13.12*	ENSTNIG0000000877	1	C	46/69/103	Pattern/Quantitative level
Fruit fly	F01	FBgn0053236	FBgn0053242	*CSNK2B*	*CSNK2B*	KMQ81491.1	2	M	19/171/172	Pattern/Quantitative level
	F02	FBgn0262610	FBgn0262575	NA	NA	FBgn0167536	1	C	20/192/196	Pattern/Quantitative level

* An outgroup sequence was obtained and used to determine the original and derived copies, but was not used for sequence analyses due to bad alignment. ** This was an exceptional case in which the shorter copy (Vmn1r213) was considered ancestral even with no outgroup, because Vmn1r214 was also the ancestral copy for the case of M21. *** The length of frameshift region/the length of the derived copy/the length of the original copy. **** “C”, ““M”, and “N” mean that the frameshift changed the reading frame in the C-terminus, middle region, or N-terminus of the amino acid sequence of a gene, respectively. ***** Showing tissue-specific expression divergence/only quantitative level expression divergence between duplicates with frameshift mutation. NA: not available.

## Data Availability

Data supporting reported results can be found in the Appendix A.

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
