# Peer review of "Functional Innovation through Gene Duplication Followed by Frameshift Mutation"

_genes, 2022, doi:10.3390/genes13020190_

Round 1

Reviewer 1 Report

The authors provide an analysis of frameshift mutations among duplicate genes to investigate their potential roles in generating novel function. I think this is an interesting question, and the authors present potentially valuable results, but I am not entirely convinced they support their claims that the frameshifts are likely adaptive. Some of my concerns are due to the lack of details presented. The authors should explain the methods so that the reader is able to reproduce these methods and results. I also could not find supplementary tables nor the supplementary dataset, only Figure S1 in the available supplementary documents.

1. The writing needs to be substantially revised for language in terms of English sentence structure, word choice, and grammar (e.g., on the first page alone: "Functionality of duplicate", "potentially importance", "have been extensively studies", etc.). This might help clarify some of my confusion presented below. In addition, please be more specific and quantitative when reporting results throughout:

"in most cases" ... what proportion of the 80 duplicate genes?
"a number of cases" ... how many?
"quite a proportion of them are well annotated" ... how many genes?

2. I expected to see a supplementary table with all the frameshift mutation information (gene names / accessions, outgroup used with ortholog accession, genomic locations of both paralogs, position of frameshift in CDS/protein, Ka/Ks, etc). The gene information would also be relevant to present in Figure 1, rather than arbitrary names. Besides communicating the essentials of your results, this will allow other researchers to reproduce your findings and conduct follow-up on your analyses, e.g., with functional assays or targeted studies.

3. From the abstract, it isn't clear here why a frameshift mutation occuring close to the C-terminus would likely be adaptive, rather than less deleterious than an N-terminus alteration. Has this been tested before? Also, would you expect adaptive frameshifts to be more common in species with higher effective population sizes and stronger effects of selection? In this study, flies were found with the fewest numbers of frameshifts among paralogs, I believe even when adjusting for genome size / gene number.

4. Methods: It isn't clear where the transcript information was used to exclude alternative splicing as a cause for the detection of frameshift mutations. Was splicing in multiple individuals and tissue types used? Some information is reported, but not enough to reproduce the results or know how extensive the sampling is.

5. Methods: It isn't clear what "confirmed with comparative genomics profile in Ensembl" means. Was each CDS mapped back to the genome to confirm the presence of unique loci for each inferred paralog? What does "manually checked with great carefulness for orthology, synteny, and sequence alignment" mean? These are important details to exclude the possibility of artifacts.

6. Methods: I would like to see more details on the support that these frameshifts are not evolutionarily relevant. Were the frameshifts cross checked with the SNP database to determine if this is possibly a rare variant? Can the authors conclude whether the frameshift is heterozygous vs homozygous in the reference, could it be allelic variation vs an independent duplicated locus? Presumably if you find the frameshift among a diverse array of transcriptomes it is common - that is why the information about the transcriptomes is important (e.g., how many individuals make up the GTEx dataset used?). Providing the genomic locations should be a minimum, but can the authors also exclude assembly or annotation errors?

7. Results: The authors argue that thei frameshift mutations are potentially functionally relevant, and conduct several analyses to support this. I would like a bit more support to exclude the possibility that these are pseudogenes. The authors showed that almost half of the genes with frameshifts have Ks_c > 0.1, and suggest these are stable rather than pseudogenes. But as the authors explain later,  some frameshifts might have 'rescued' pseudogenes, so one might assume the opposite could also occur - functional (adaptive) divergence of duplicates followed by recent pseudogenization, either via frameshift or some other mechanism, thereby resulting in Ks_c > 0.1. 
It isn't clear to me what Ks_c is expected for pseudogenes, can the authors show this with existing/known pseudogenes? 

8. Results: I need help to understand the methods and interpretation of the Ka/Ks of frameshifted regions - to calculate Ks/Ks properly, you would align a shifted frame with an unshifted frame, whereby most sites will differ except among repetitive regions. Doesn't this change the mutational expectations in such a way that it no longer meets the assumptions for substitutions and a Ka/Ks calculation? The authors mention they estimated the ancestral sequence for each pair (I don't know how they derived this), but once there is a frameshift, subsequent substitutions should be compared to the frameshifted ancestor, not the pre-frameshifted ancestor. Providing information on how the ancestral sequence was estimated, and showing the actual alignments would help evaluate the approach. Without this context, I do not know how to interpret these results and whether they are valid.

9.I'm wondering if additional analyses might clarify some of the concerns stated above:
*How many frameshift mutations are there among non-paralogs? Can the authors calculte the frequency of frameshift mutations between orthologs to show whether duplication potentially increases this phenomenon?
*How certain are we that the outgroups represent the ancestral copy? Support from additional outgroups would help prevent misidentifying which is the derived copy. Is the conserved copy expressed similarly to the outgroup ortholog as we would expect? There was very little reported on the 'significant divergence in expression' of the duplicate copies, which I think would be useful to present, especially if the outgroup expression could be presented to show whether the derived copy also has the most diverged expression pattern.
*Can the authors compare the Ka/Ks in recent pseudogenes with frameshift mutations as a method to support an increase in Ka/Ks among their paralogs?

Minor comments
I found it weird that the authors did not report the NOTCH genes among their frameshift mutations, wouldn't we expect this to be identified? It would provide as a positive control for their methods.
It isn't clear what "various patterns of modifications of the reading frame" means
Can you revise "...have non-negligible numbers of genes that have experienced frameshift mutations, which evolved through gene duplication."? it can be interpreted that the frameshift itself is duplicated, rather than gene duplication that allowed the evolution of the mutation, e.g., under relaxed selection 
It would be informative to present results on the 'significant divergence in expression' and the associated statistics in the text and/or in a figure, rather than in Table S1 (which I could not find to download).

Author Response

Reviewer 1

The authors provide an analysis of frameshift mutations among duplicate genes to investigate their potential roles in generating novel function. I think this is an interesting question, and the authors present potentially valuable results, but I am not entirely convinced they support their claims that the frameshifts are likely adaptive. Some of my concerns are due to the lack of details presented. The authors should explain the methods so that the reader is able to reproduce these methods and results. I also could not find supplementary tables nor the supplementary dataset, only Figure S1 in the available supplementary documents.

We have provided more details of methods and supplementary tables in the revision.

  1. The writing needs to be substantially revised for language in terms of English sentence structure, word choice, and grammar (e.g., on the first page alone: "Functionality of duplicate", "potentially importance", "have been extensively studies", etc.). This might help clarify some of my confusion presented below. In addition, please be more specific and quantitative when reporting results throughout:

"in most cases" ... what proportion of the 80 duplicate genes?
"a number of cases" ... how many?
"quite a proportion of them are well annotated" ... how many genes?

We have revised the writing accordingly and presented results quantitatively not only in the main but also in the abstract.

  1. I expected to see a supplementary table with all the frameshift mutation information (gene names / accessions, outgroup used with ortholog accession, genomic locations of both paralogs, position of frameshift in CDS/protein, Ka/Ks, etc). The gene information would also be relevant to present in Figure 1, rather than arbitrary names. Besides communicating the essentials of your results, this will allow other researchers to reproduce your findings and conduct follow-up on your analyses, e.g., with functional assays or targeted studies.

Yes, all information the reviewer expected was listed in table S1 and S2.

  1. From the abstract, it isn't clear here why a frameshift mutation occurring close to the C-terminus would likely be adaptive, rather than less deleterious than an N-terminus alteration. Has this been tested before? Also, would you expect adaptive frameshifts to be more common in species with higher effective population sizes and stronger effects of selection? In this study, flies were found with the fewest numbers of frameshifts among paralogs, I believe even when adjusting for genome size / gene number.

We agreed on the reviewer’s point on whether duplicate with frameshifts is adaptive. If so, we might observe more such cases in species with larger effective population size, e.g., more cases in fly than human after genome size / gene adjustment. However, we have not claimed that duplicates with frameshifts are adaptive because our KaKs calculation do not detect obviously adaptive signal in our data. Therefore, we argued that duplicate with frameshifts may likely contribute to adaptive evolution based on the obvious reported cases of NOTCH and ARHGAP11genes.

  1. Methods: It isn't clear where the transcript information was used to exclude alternative splicing as a cause for the detection of frameshift mutations. Was splicing in multiple individuals and tissue types used? Some information is reported, but not enough to reproduce the results or know how extensive the sampling is.

Yes, we considered all transcripts for each gene to exclude frameshift that was introduced by alternative splicing. We have now added the details in the Materials and Methods section in the revision.

  1. Methods: It isn't clear what "confirmed with comparative genomics profile in Ensembl" means. Was each CDS mapped back to the genome to confirm the presence of unique loci for each inferred paralog? What does "manually checked with great carefulness for orthology, synteny, and sequence alignment" mean? These are important details to exclude the possibility of artifacts.

"…confirmed with comparative genomics profile in Ensembl" and "…manually checked with great carefulness for orthology, synteny, and sequence alignment" refer to that we used the multiple sequence alignments of orthologs and paralogs in Ensembl to confirm our identification of duplicate with frameshift mutations. We added this notification in the text, see page 7 the last sentence in the second paragraph in the text.

  1. Methods: I would like to see more details on the support that these frameshifts are not evolutionarily relevant. Were the frameshifts cross checked with the SNP database to determine if this is possibly a rare variant? Can the authors conclude whether the frameshift is heterozygous vs homozygous in the reference, could it be allelic variation vs an independent duplicated locus? Presumably if you find the frameshift among a diverse array of transcriptomes it is common - that is why the information about the transcriptomes is important (e.g., how many individuals make up the GTEx dataset used?). Providing the genomic locations should be a minimum, but can the authors also exclude assembly or annotation errors?

We did not use SNP database to confirm our identification of duplicate with frameshift mutations. We tried but unfortunately polymorphism data were not available for most of those duplicated genes perhaps because of technical difficulties.

  1. Results: The authors argue that the frameshift mutations are potentially functionally relevant, and conduct several analyses to support this. I would like a bit more support to exclude the possibility that these are pseudogenes. The authors showed that almost half of the genes with frameshifts have Ks_c > 0.1, and suggest these are stable rather than pseudogenes. But as the authors explain later, some frameshifts might have 'rescued' pseudogenes, so one might assume the opposite could also occur - functional (adaptive) divergence of duplicates followed by recent pseudogenization, either via frameshift or some other mechanism, thereby resulting in Ks_c > 0.1. 
    It isn't clear to me what Ks_c is expected for pseudogenes, can the authors show this with existing/known pseudogenes? 

The prediction is simple: selection is relaxed completely after pseudogenization, so that Ka and Ks evolve at similar rates. Sorry for the confusion, we should have said Ks_c is fairly large “while maintaining Ka_c low”. See page 5 line 4-5 in the first paragraph.

  1. Results: I need help to understand the methods and interpretation of the Ka/Ks of frameshifted regions - to calculate Ks/Ks properly, you would align a shifted frame with an unshifted frame, whereby most sites will differ except among repetitive regions. Doesn't this change the mutational expectations in such a way that it no longer meets the assumptions for substitutions and a Ka/Ks calculation? The authors mention they estimated the ancestral sequence for each pair (I don't know how they derived this), but once there is a frameshift, subsequent substitutions should be compared to the frameshifted ancestor, not the pre-frameshifted ancestor. Providing information on how the ancestral sequence was estimated, and showing the actual alignments would help evaluate the approach. Without this context, I do not know how to interpret these results and whether they are valid.

In fact, the sequence of duplicate with frameshift mutations we identified are very conserved at nucleotide level. We thus first changed the nucleotide sequence of duplicate with frameshift mutations back to its original reading frame by adjusting the frameshift mutation. Then, ancestry sequence was first reconstructed based on duplicates and their closely related outgroup sequence at nucleotide level. Finally, selection pressure was evaluated by calculating Ka and Ks in each copy of duplicates with comparison between duplicate sequence and its ancestry sequence for common frame region and the frameshifted region, respectively. Specially, Ka and Ks in the derived copy with frameshift mutation in the frameshifted region were calculated with its true reading frame by adjusting reading frame of its ancestry sequence. We added those details in the text, see page 5 line 10 in the second paragraph and page 7 line 10-15 in the third paragraph.

  1. I'm wondering if additional analyses might clarify some of the concerns stated above:
    How many frameshift mutations are there among non-paralogs? Can the authors calculte the frequency of frameshift mutations between orthologs to show whether duplication potentially increases this phenomenon?
    How certain are we that the outgroups represent the ancestral copy? Support from additional outgroups would help prevent misidentifying which is the derived copy. Is the conserved copy expressed similarly to the outgroup ortholog as we would expect? There was very little reported on the 'significant divergence in expression' of the duplicate copies, which I think would be useful to present, especially if the outgroup expression could be presented to show whether the derived copy also has the most diverged expression pattern.
    Can the authors compare the Ka/Ks in recent pseudogenes with frameshift mutations as a method to support an increase in Ka/Ks among their paralogs?

Thanks for all those constructive suggestions. We would have considered those constructive suggestions, viz. frequency of frameshift occurrence among orthologs, comparison with the most closely related outgroup sequence and expression pattern, and comparison of Ka/Ks with recent pseudogenes in our further studies. In this study, we chose to aim on the identification of duplicate with frameshift mutations based on sequence analysis.

Minor comments
I found it weird that the authors did not report the NOTCH genes among their frameshift mutations, wouldn't we expect this to be identified? It would provide as a positive control for their methods.

Our work is inspired by the NOTCH gene studies. The current human genome annotation has not updated for NOTCH genes. Thus, it is not surprised that we did not report NOTCH genes in our study. But ARHGAP11 is a clear case as the reviewer expected for NOTCH genes. We added explanations for uncovering NOTCH genes in the text, see page 2 line 7-8 in the third paragraph.

It isn't clear what "various patterns of modifications of the reading frame" means.

It means the modifications of the reading frame involves type-C, -N, and/or -M as showed in Figure 1.

Can you revise "...have non-negligible numbers of genes that have experienced frameshift mutations, which evolved through gene duplication."? it can be interpreted that the frameshift itself is duplicated, rather than gene duplication that allowed the evolution of the mutation, e.g., under relaxed selection.

We have written the sentence by following the reviewer’s suggestion.

It would be informative to present results on the 'significant divergence in expression' and the associated statistics in the text and/or in a figure, rather than in Table S1 (which I could not find to download).

The divergence in quantitative expression has not been done exhaustively. Thus, we have revised the writing and toned down the claim.

Reviewer 2 Report

The manuscript “Frequently functional innovation through gene duplication followed by frameshift mutation” by Guo and colleagues is a study that set out to identify duplicated genes with frameshift mutation as potential candidates for neofunctionalization in humans, mice,  zebrafish, and fruit-flies. In total 80 genes were identified. It was very nice that the authors double checked orthology of the genes to help determine which gene variant was most likely ancestral and which one most likely derived. It was less fortunate that the authors kept a very global look at their results without exploring the potential of neofunctionalization. The rational of the various tests performed was not very clear and it was completely unclear what controls were used. Overall, this exploratory study has potential if a) the results are more clearly and cleanly described, including what genes are actually studied, and b) the results are discussed with various explanations in mind. The authors only considered amino acid mutations as an important factor and not how and when genes were expressed. Below are additional concerns that need to be addressed.

Major concerns

+ The authors set out to study four commonly studies species, which have diverged ~800 MYA. Is it the goal of the authors to identify ancient duplication and their speculated neofunctionalization or duplication event that more recently were involved in neofunctionalization? Based on the examples described in the preceding paragraphs, the reader might expect the latter. This is also important in order to know how old the frameshift mutations are. If the occurred recently, limited time is available for the accumulation of additional mutations, one would expect if two diverged duplicated genes have distinct functions (as a consequence of neofunctionalization).

+ To determine age of duplication, the authors use Ks_C from the 80 gene pairs identified. This seems like a small number of genes to determine age, plus these genes were selected for their distinct history of gene duplication and neofunctionalization.

+ Overall, little detail is provided about what is the actual findings are beyond global statements. As four species were assessed, what was unique about each of them from the perspective of the frameshift mutations?

+ Figure 1 is confusing. Do the numbers above each original and derived bar represent residue number? What genes are represented? From species are these?

+ It might be worth making a table of the 53 confirmed gene pairs to provide basic information (gene name, db code, size, (predicted) function), as these genes might be of specific interest to other researchers. Why hide it in the supplemental?

+ What outgroups specifically were used? These not mentioned anywhere in the text.

+ The authors tried to estimate the ancestral sequence for each pair of duplicates, but no results are reported or discussed.

+ The authors describe increased, yet statistically not significant, substitution in the frameshift region. This section is very confusing. Maybe it is worth rewriting this section focusing on clearly explaining what was done, what was found, and what it means.

+ If neofunctionalization did happen, following a frameshift mutation, wouldn’t it be expected that the substation rate increased not only in the frameshift region, but also outside of it. Or do the authors think that a different promoter is driving a different expression pattern that is driving the neofunctionalization? There is plenty of RNA-seq data available where the expression pattern of these frameshift genes could be assessed to predict neofunctionalization.

+ The authors state, on page 5, that pseudogenization occurs shortly after duplication, but a reference is missing.

+ In the discussion, the authors speculate how a type-N frameshift mutation could produce a pseudogene, which than acquired a secondary frameshift mutation to produce a functional gene again. This speculated scenario is then followed up by a statement that several of such events were found by the authors. This is a puzzling series of statements. One could speculate and predict that some of the results found could be explained by this speculated scenario. Ideally, the authors would also suggest how such a scenario could be proven.

+ The final concluding paragraph is very vague. Not at any point have the authors shown that neofunctionalization has occurred, as postulated by Ohno. Identifying candidate genes is important, but this should be clearly stated as such.

+ Tables S1 and S2 were not provided.

Minor concerns

+ On page 1,  the authors mention the duplication event of NOTCH genes and that NOTCH2 was formed after the split with chimpanzee, whereas Fiddes et al 2018 showed that NOTCH duplication occurred after the split of the orangutans and the other ape lineages.

+ Page 2, second line: “presumably” should be “is speculated to have”.

+ For instance, on page 2, last. Line of the third paragraph, the. authors reference supplemental data, but do not specify which specific supplemental data. When referencing figures or tables, please be specific to make it easier for the reader to follow.

+ Figure 2 – how were bin-sizes determined?

+ Headers for Figure 2E and 2F are identical.

+ Figures 2E and 2F should be showing relative number per species rather than absolute. It would also be worthwhile if the authors could speculate on why the distribution of relative Ka/Ks numbers differ between species. Does this relate to their respective mutation rates?

Author Response

The manuscript “Frequently functional innovation through gene duplication followed by frameshift mutation” by Guo and colleagues is a study that set out to identify duplicated genes with frameshift mutation as potential candidates for neofunctionalization in humans, mice, zebrafish, and fruit-flies. In total 80 genes were identified. It was very nice that the authors double checked orthology of the genes to help determine which gene variant was most likely ancestral and which one most likely derived. It was less fortunate that the authors kept a very global look at their results without exploring the potential of neofunctionalization. The rational of the various tests performed was not very clear and it was completely unclear what controls were used. Overall, this exploratory study has potential if a) the results are more clearly and cleanly described, including what genes are actually studied, and b) the results are discussed with various explanations in mind. The authors only considered amino acid mutations as an important factor and not how and when genes were expressed. Below are additional concerns that need to be addressed. 

We have provided more details of methods and supplementary tables in the revision. The aim of the study is mainly from the sequence perspective to show that genomes have non-negligible numbers of genes that have experienced frameshift mutations, which evolved through gene duplication.

Major concerns

The authors set out to study four commonly studies species, which have diverged ~800 MYA. Is it the goal of the authors to identify ancient duplication and their speculated neofunctionalization or duplication event that more recently were involved in neofunctionalization? Based on the examples described in the preceding paragraphs, the reader might expect the latter. This is also important in order to know how old the frameshift mutations are. If the occurred recently, limited time is available for the accumulation of additional mutations, one would expect if two diverged duplicated genes have distinct functions (as a consequence of neofunctionalization).

We aimed to identify duplicates with frameshifts in well-annotated genomes, regardless of if duplicates are ancient or young. To this end, we found that

To determine age of duplication, the authors use Ks_C from the 80 gene pairs identified. This seems like a small number of genes to determine age, plus these genes were selected for their distinct history of gene duplication and neofunctionalization.

Yes, we only determine the age of each of the 80 duplicate pairs with Ks_C.

Overall, little detail is provided about what is the actual findings are beyond global statements. As four species were assessed, what was unique about each of them from the perspective of the frameshift mutations?

In fact, the patterns in the studied species are similar, let’s take the frameshift types (-C, -N, and -M) (Figure 2D) as an example.

Figure 1 is confusing. Do the numbers above each original and derived bar represent residue number? What genes are represented? From species are these?

We have added those details in the legend.

It might be worth making a table of the 53 confirmed gene pairs to provide basic information (gene name, db code, size, (predicted) function), as these genes might be of specific interest to other researchers. Why hide it in the supplemental?

We have listed gene information in table S1 and S2. We do not mean to hide such information. In fact, we uploaded table S1 and S2 to the submission system but have no idea why they could not been seen by reviewers.

What outgroups specifically were used? These not mentioned anywhere in the text.

All outgroup information is listed in table S1.

The authors tried to estimate the ancestral sequence for each pair of duplicates, but no results are reported or discussed. The authors describe increased, yet statistically not significant, substitution in the frameshift region. This section is very confusing. Maybe it is worth rewriting this section focusing on clearly explaining what was done, what was found, and what it means.

We have now added details in Materials and Methods section in the revision, “Specially, to estimate the selection pressure, ancestry sequence was first reconstructed based on duplicates and their closely related outgroup sequence. Then, Ka and Ks in each copy of duplicates were calculated by comparison between duplicate sequence and its ancestry sequence for common frame region and the frameshifted region, respectively.”

If neofunctionalization did happen, following a frameshift mutation, wouldn’t it be expected that the substation rate increased not only in the frameshift region, but also outside of it. Or do the authors think that a different promoter is driving a different expression pattern that is driving the neofunctionalization? There is plenty of RNA-seq data available where the expression pattern of these frameshift genes could be assessed to predict neofunctionalization.

The cases of NOTCH and ARHGAP11 genes do not support the scenario that substitution rate increased not only in the frameshift region but also outside of it, which suggests strong selection pressure on the duplicates after frameshift mutation. It is true that RNA-seq data would be helpful for complementary studies. We will take it into account in our following studies and also wish to see others will confirm our findings with RNA-seq data.

The authors state, on page 5, that pseudogenization occurs shortly after duplication, but a reference is missing.

Added.

In the discussion, the authors speculate how a type-N frameshift mutation could produce a pseudogene, which than acquired a secondary frameshift mutation to produce a functional gene again. This speculated scenario is then followed up by a statement that several of such events were found by the authors. This is a puzzling series of statements. One could speculate and predict that some of the results found could be explained by this speculated scenario. Ideally, the authors would also suggest how such a scenario could be proven.

We rephrased the corresponding part by saying “one possible scenario is that …”. See page 6 line 9 in the third paragraph.

The final concluding paragraph is very vague. Not at any point have the authors shown that neofunctionalization has occurred, as postulated by Ohno. Identifying candidate genes is important, but this should be clearly stated as such.

We have added the NOTCH and ARHGAP11 genes to clearly state the significance of gene duplication followed by frameshift mutation.

Tables S1 and S2 were not provided.

Table S1 and S2 are provided now.

Minor concerns

On page 1, the authors mention the duplication event of NOTCH genes and that NOTCH2 was formed after the split with chimpanzee, whereas Fiddes et al 2018 showed that NOTCH duplication occurred after the split of the orangutans and the other ape lineages.

It is true that NOTCH duplication occurred after the split of the orangutans and the other ape lineages, but a copy of human NOTCH2 genes acquired a new function by a 4 bp deletion that introduced a fragment of completely new amino acid sequences after the split with chimpanzee.

Page 2, second line: “presumably” should be “is speculated to have”.

Done.

For instance, on page 2, last. Line of the third paragraph, the. authors reference supplemental data, but do not specify which specific supplemental data. When referencing figures or tables, please be specific to make it easier for the reader to follow.

We have followed the suggestion and revised the MS accordingly.

Figure 2 – how were bin-sizes determined?

The bin size is 0.1. We mentioned in the legend.

Headers for Figure 2E and 2F are identical.

The header for figure 2E has been correct.

Figures 2E and 2F should be showing relative number per species rather than absolute. It would also be worthwhile if the authors could speculate on why the distribution of relative Ka/Ks numbers differ between species. Does this relate to their respective mutation rates?

We do not think they are significantly different, rather they are quite similar to each other.

Reviewer 3 Report

The authors have searched for paralogs with frameshift mutations and they test the hypothesis that these could result in neofunctionalization. Although they cannot provide clear evidence for neofunctionalization, the study is well-designed and performed and they provide interesting results.

I give some specific comments below:

-I think they authors should provide the outgroups that they have used for each genome. Did they check if the duplication occurred before the split between the referred genome and the outgroup?

- It is not clear between which sequences they have calculated Ka/Ks. Between the two paralogs or between each paralog with the outgroup? When they estimate the relative time of duplication from Ks-c it seems that they compare the two paralogs. For the Kc_d it seems that they use each paralog with the outgroup.

-“A relatively large divergence between paralogs (i.e., Ks_c > 0.1) should indicates that these derived

copies have been preserved as functional genes for a significant amount of time.” It might also mean that the duplication event occurred earlier in time. (by the way it should be “indicate” without the final “s”).

-In the paragraph “Figures 2C and D show” why is there a threshold of 0.3 in the Ks?

- In the next paragraph the authors mention that they have found the know duplication in the ARHGAP gene. I think they should provide evidence whether they identify other know duplication events (e.g. the NOTCH duplication) not only in humans but in other organisms as well. This will be a control that their method works well and does not miss duplication events.  

- It was not clear how they ruled out the possibility of gene conversion. Gene conversion can be a major source of error when comparing paralogs.

- It should be useful to give the genomic distance between the paralogs.

-The abstract needs to be written in a simpler and clearer way. For example the sentence “The frameshift mutation preferentially locates close to the C-terminus in most cases, which indicates that a frameshift mutation that change the reading frame in a small part may likely contribute to adaptive evolution while maintaining the major part of the coding gene, otherwise too deleterious to survive.” Is not very clear.

-There are several typos. I have already mentioned a typo above, in the abstract “role of frameshift mutationS in creating functional novelty.”, in the 1st paragraph of introduction “have been extensively studiED in a wide range of species” and others.

Author Response

The authors have searched for paralogs with frameshift mutations and they test the hypothesis that these could result in neofunctionalization. Although they cannot provide clear evidence for neofunctionalization, the study is well-designed and performed and they provide interesting results.

Thanks for the acknowledgement of our study.

I give some specific comments below:

I think they authors should provide the outgroups that they have used for each genome. Did they check if the duplication occurred before the split between the referred genome and the outgroup?

We have provided such information in table S1 and S2, if closely related outgroups in Ensembl or NCBI are available for a certain duplicate pair. It says that duplication occurred after the split with the outgroup we used.

It is not clear between which sequences they have calculated Ka/Ks. Between the two paralogs or between each paralog with the outgroup? When they estimate the relative time of duplication from Ks-c it seems that they compare the two paralogs. For the Kc_d it seems that they use each paralog with the outgroup.

When we estimated the relative time of duplication based on Ks_C, we calculated Ka/Ks between duplicates (the two paralogs). When we estimated the selection pressure, we calculated Ka/Ks between each copy of duplicates and ancestry sequence that was reconstructed based on duplicates (the two paralogs) and the outgroup in common frame region and in the frameshifted region, respectively. We have now added the details in Materials and Methods section in the revision.

“A relatively large divergence between paralogs (i.e., Ks_c > 0.1) should indicates that these derived copies have been preserved as functional genes for a significant amount of time.” It might also mean that the duplication event occurred earlier in time. (by the way it should be “indicate” without the final “s”).

We agree with the reviewer on this and correct the typo.

In the paragraph “Figures 2C and D show” why is there a threshold of 0.3 in the Ks?

We did not set a threshold. It is merely due to that 73 out of 80 cases have that Ks_C < 0.3.

In the next paragraph the authors mention that they have found the know duplication in the ARHGAP gene. I think they should provide evidence whether they identify other know duplication events (e.g. the NOTCH duplication) not only in humans but in other organisms as well. This will be a control that their method works well and does not miss duplication events.  

Yes, we provided several cases in other species with reports from earlier studies. However, NOTCH genes are not identified by our study since the current human genome annotation has not updated for NOTCH genes, see our reply to the reviewer #1. We also mentioned this in the text, see page 2 line 7-8 in the third paragraph.

It was not clear how they ruled out the possibility of gene conversion. Gene conversion can be a major source of error when comparing paralogs.

We do not have to rule out the possibility of gene conversion. It merely retards divergence, thereby causing an underestimation of ages as was mentioned in the text (see page 2, line 15-16 in the third paragraph).

It should be useful to give the genomic distance between the paralogs.

We have listed gene information in table S1, and most of duplicates are located on different chromosomes in the four genomes we studied.

The abstract needs to be written in a simpler and clearer way. For example the sentence “The frameshift mutation preferentially locates close to the C-terminus in most cases, which indicates that a frameshift mutation that change the reading frame in a small part may likely contribute to adaptive evolution while maintaining the major part of the coding gene, otherwise too deleterious to survive.” Is not very clear.

We have rewritten this sentence.

There are several typos. I have already mentioned a typo above, in the abstract “role of frameshift mutationS in creating functional novelty.”, in the 1st paragraph of introduction “have been extensively studiED in a wide range of species” and others.

We have revised and corrected typos accordingly.

Round 2

Reviewer 1 Report

Some improvements and clarifications were made, but most of my questions and concerns remain despite the revisions. Below I've numbered my comments in the same order as my previous ones - most were insufficiently addressed.

1.There remains grammatical and structural errors that the authors did not address, including two that I previously indicated from the abstract. 
-Change to "demonstrate the potential importance"

In addition, here are more on the second page
-'thereby motivated us to explore'  should be motivating
-'increasing basal progenitors' does not make sense or needs context
-'in the adaptive evolution' should be 'involved in adaptive evolution'
-'were reported' should be 'have been reported'
-'and almost none were convincing cases of Ohno’s prediction except for one case' - I don't know what the authors mean by 'convincing', which is subjective. Also, do the authors mean 'a single case'? Otherwise it is confusing to state 'almost none' followed by 'except for one'
-'Thus, in animals,' sentence is redundant with the first sentence of the paragraph.
-The sentence starting with "We here report" should be made into two sentences. It isn't clear if the 'diverse organisms' refers to the 4 species they mention, or additional ones.
-'per site in common frame region' should be 'per site in common frame regions' plural
-"We then searched for an outgroups sequence in a species such that" can be changed to "We then searched for an outgroup sequence such that"
-"We successfully determined" should be "We determined"
There are plenty more errors, the authors should get the language checked. I had to re-read many sentences to try to (sometimes unsuccessfully) understand their meaning.

2. Supplementary Tables need work. Provide explanations for each column, I can't tell what some of the columns mean.
Refer to "type-M" in the main text, and define the types here
Why do some have Ka_C/Ks_C have NA when they have both values in the previous columns?
Why do some have Outgroup "No" but have an outgroup ID?
Expression column is lacking detail, why is only 'quantitative level' or 'pattern' provided? 
Some gene IDs include more than one ID, why?
Some outgroup IDs are genes, others are transcripts, why?

Change 'bad alignment' to describe what you mean by 'bad alignment' - how was good vs bad assessed? 
Outgrpup should be outgroup
Orifibal should be original

3. Abstract statements need better context.
"Functionality of duplicate involving frameshift mutations was confirmed by expression and DNA sequences". Do they mean "of duplicate genes", and "suggesting a potential role"? And what is meant by 'confirmed by. ... DNA sequences', do they mean patterns of substitutions in the gene outside the frameshift?

Errors in the sentence:
"The frameshift mutation preferentially locates close to the C-terminus in most cases (55/88), which indicates that a frameshift mutation that change the reading frame in a small part at the end of a duplicate may likely contribute to adaptive evolution, otherwise too deleterious to survive."

*The authors need to justify this statement that having c-terminus frameshifts mean that they likely are adaptive. And why is the alternative 'too deleterious', can it not be near neutral / slightly deleterious? In one of the responses, the authors mention that this statement is based on two examples (NOTCH and ARHGAP11), which is a very small biased sample size, and not based on their own results.

The next sentence starts with "However", but the authors don't make it clear why they use 'however' - is it because they mean that the 11/80 are likely deleterious?

4. The authors still need to do a better job explaining what they did and how they evaluated alternative splicing. I do not follow what the authors mean by:
"It means that all transcripts for each gene to exclude frameshift that is introduced by alternative splicing."

5. The authors stop short at explaining what they did to confirm the frameshift mutations. They added that they use multiple sequence alignments:  "Finally, the candidate duplicated pairs with frameshift mutations were manually checked with great carefulness for orthology, synteny, and sequence alignment by utilizing multiple sequence alignments of orthologs and paralogs in Ensembl."
Can the authors explain how the multiple sequence alignments helped confirm frameshift mutations? Were all orthologs from outgroup species utilized in the MSA, to make sure that the frameshift was uniquely found in the focal species? It would be important to consider all duplicates from all species (including outparalogs).
Can the authors explain how synteny was assessed? For example, by making sure that adjacent genes in outgroups were orthologs to at least one of the duplicate pair?
It still isn't clear what "confirmed with comparative genomics profile in Ensembl" refers to, and how that can confirm frameshifts (other than determining there is a frameshift in the reference assembly). What you need are independent sources of evidence for confirmation.

6. Since the authors seem to have a single individual (the reference genomes) with these frameshifts, they cannot exclude the possibility of annotation/assembly errors. Unless they can exclude this possibility and detect the presence of the frameshift in another individual, I expect to see a clear caveat statement acknowledging this.

7. A recent frameshift mutation would also show this pattern - the duplicate genes with high Ks_c and low Ka_c recently encountered a non-adaptive frameshift mutation (with insufficient time to elevate Ka_c). Is the assumption here that the observed frameshift mutations are fixed within the respective species (and thus not a recent non-adaptive frameshift)?

It isn't clear how 'expression divergence' was assessed, only that some statistics were carried out in R. Just because expression has diverged does not mean they are both functional.

It isn't clear whether the authors isolated RNA-seq reads overlapping the frame shifted regions in case alternative splicing occurs that excludes that portion of the gene (and therefore does not include a transcript with a frameshift)

Please revise the sentence in the abstract and end of discussion to this: "have non-negligible numbers of genes that have experienced frameshift mutations following gene duplication." 

Given the uncertainties I still have and the inability to add analyses, I'm not convinced that these frameshift mutations are adaptive, nor that they provide novel function.  The Ka_DF/Ks_DF analysis suggests that when pooled, the frame shifted region evolves neutrally. In the least, the authors need to be more careful with strong statements such as "on a genomic scale we observed a number of cases that have been benefited from frameshift mutations".

Author Response

see our replies

Reviewer 2 Report

Overall, the authors provided an answer to each point raised by this reviewer. It would have been very helpful if the altered text in the updated manuscript was in a different color from the original text, so it would be easier for the reviewer to see what changed. Consequently, it is unclear to this reviewer if the first major concern raised was actually addressed in the text (aside from the fact that the answer seems to end in the middle of a sentence). In fact, there appears to be no clarification in the text about the first point brought up. The overall answers provided by the authors is very brisk. Therefore, this reviewer has addressed each answer individually (see below ++).

The manuscript “Frequently functional innovation through gene duplication followed by frameshift mutation” by Guo and colleagues is a study that set out to identify duplicated genes with frameshift mutation as potential candidates for neofunctionalization in humans, mice, zebrafish, and fruit-flies. In total 80 genes were identified. It was very nice that the authors double checked orthology of the genes to help determine which gene variant was most likely ancestral and which one most likely derived. It was less fortunate that the authors kept a very global look at their results without exploring the potential of neofunctionalization. The rational of the various tests performed was not very clear and it was completely unclear what controls were used. Overall, this exploratory study has potential if a) the results are more clearly and cleanly described, including what genes are actually studied, and b) the results are discussed with various explanations in mind. The authors only considered amino acid mutations as an important factor and not how and when genes were expressed. Below are additional concerns that need to be addressed. 

# We have provided more details of methods and supplementary tables in the revision. The aim of the study is mainly from the sequence perspective to show that genomes have non-negligible numbers of genes that have experienced frameshift mutations, which evolved through gene duplication.

++ The concern that the rational of the various tests performed was not clear has not been addressed, as this can only be addressed in the text.

Major concerns

+ The authors set out to study four commonly studies species, which have diverged ~800 MYA. Is it the goal of the authors to identify ancient duplication and their speculated neofunctionalization or duplication event that more recently were involved in neofunctionalization? Based on the examples described in the preceding paragraphs, the reader might expect the latter. This is also important in order to know how old the frameshift mutations are. If the occurred recently, limited time is available for the accumulation of additional mutations, one would expect if two diverged duplicated genes have distinct functions (as a consequence of neofunctionalization).

# We aimed to identify duplicates with frameshifts in well-annotated genomes, regardless of if duplicates are ancient or young. To this end, we found that

++ On top of the incomplete answer, the concern is not addressed. The age of a duplicated gene and their potential neofunctionalization are linked. In fact, to better identify gene duplication events, using an outgroup for each species of interest would be very helpful as it would prune out any older duplicated genes. As for neofunctionalization, this of course would be speculative, unless others have already shown this for a specific gene. This concern must be adequately addressed, either by using the outgroups (ideal situation) or rewriting the text to clearly state the caveats and assumptions that limit the scope of this study.

+ To determine age of duplication, the authors use Ks_C from the 80 gene pairs identified. This seems like a small number of genes to determine age, plus these genes were selected for their distinct history of gene duplication and neofunctionalization.

# Yes, we only determine the age of each of the 80 duplicate pairs with Ks_C.

++ Thus the authors are not concerned about the implications of a small sample size to determine age. This potential caveat must be clearly noted in the text.

+ Overall, little detail is provided about what is the actual findings are beyond global statements. As four species were assessed, what was unique about each of them from the perspective of the frameshift mutations?

# In fact, the patterns in the studied species are similar, let’s take the frameshift types (-C, -N, and -M) (Figure 2D) as an example.

++ This is an incomplete answer and does not address the concern.

+ Figure 1 is confusing. Do the numbers above each original and derived bar represent residue number? What genes are represented? From species are these?

# We have added those details in the legend.

++ Thank you for updating and clarifying Figure 1.

+ It might be worth making a table of the 53 confirmed gene pairs to provide basic information (gene name, db code, size, (predicted) function), as these genes might be of specific interest to other researchers. Why hide it in the supplemental?

# We have listed gene information in table S1 and S2. We do not mean to hide such information. In fact, we uploaded table S1 and S2 to the submission system but have no idea why they could not been seen by reviewers.

++ It is indeed strange that this reviewer cannot see the supplementary tables. Presumably the editor can clarify this.

+ What outgroups specifically were used? These not mentioned anywhere in the text.

# All outgroup information is listed in table S1.

++ Important information as outgroup should not be hidden in a supplementary table but must be clearly stated in the text. Please do so.

+ The authors tried to estimate the ancestral sequence for each pair of duplicates, but no results are reported or discussed. The authors describe increased, yet statistically not significant, substitution in the frameshift region. This section is very confusing. Maybe it is worth rewriting this section focusing on clearly explaining what was done, what was found, and what it means.

# We have now added details in Materials and Methods section in the revision, “Specially, to estimate the selection pressure, ancestry sequence was first reconstructed based on duplicates and their closely related outgroup sequence. Then, Ka and Ks in each copy of duplicates were calculated by comparison between duplicate sequence and its ancestry sequence for common frame region and the frameshifted region, respectively.”

++ It is wonderful to see that the methods section is updated. Ideally, also the main text is updated to clearly report the findings.

+ If neofunctionalization did happen, following a frameshift mutation, wouldn’t it be expected that the substation rate increased not only in the frameshift region, but also outside of it. Or do the authors think that a different promoter is driving a different expression pattern that is driving the neofunctionalization? There is plenty of RNA-seq data available where the expression pattern of these frameshift genes could be assessed to predict neofunctionalization.

# The cases of NOTCH and ARHGAP11 genes do not support the scenario that substitution rate increased not only in the frameshift region but also outside of it, which suggests strong selection pressure on the duplicates after frameshift mutation. It is true that RNA-seq data would be helpful for complementary studies. We will take it into account in our following studies and also wish to see others will confirm our findings with RNA-seq data.

++ Thank you for this answer. This reviewer would indeed expect a change in selection pressure after a frame-shift mutation as any functional region that was lost will alter the total functional role of that frame-shift duplicated gene. Either it takes up a completely new role for which is has not undergone selection or becomes optimized to one of the several functions that the original gene performed. If neutral or near-neutral evolution is expected to prevail, it would be informative if the authors could articulate in the manuscript how and why this should happen. Alternatively, the authors should expand the paragraph on the potential evolutionary path of frameshift duplicated genes, beyond the suggestion that a frame-shift duplicated gene first becomes a pseudogene before it becomes functional again.

+ The authors state, on page 5, that pseudogenization occurs shortly after duplication, but a reference is missing.

# Added.

++ Thank you.

+ In the discussion, the authors speculate how a type-N frameshift mutation could produce a pseudogene, which than acquired a secondary frameshift mutation to produce a functional gene again. This speculated scenario is then followed up by a statement that several of such events were found by the authors. This is a puzzling series of statements. One could speculate and predict that some of the results found could be explained by this speculated scenario. Ideally, the authors would also suggest how such a scenario could be proven.

# We rephrased the corresponding part by saying “one possible scenario is that …”. See page 6 line 9 in the third paragraph.

++ Thank you. It would also be ideal if the authors could address the last part of this concern.

+ The final concluding paragraph is very vague. Not at any point have the authors shown that neofunctionalization has occurred, as postulated by Ohno. Identifying candidate genes is important, but this should be clearly stated as such.

# We have added the NOTCH and ARHGAP11 genes to clearly state the significance of gene duplication followed by frameshift mutation.

++ It is very helpful to add two examples from the literature. It should now be clear to the reader that this study aims to identify frameshift duplicated genes for which neofunctionalization has not been shown.

+ Tables S1 and S2 were not provided.

# Table S1 and S2 are provided now.

++ This reviewer still can’t see them. The editor will have to correct this.

Minor concerns

+ On page 1, the authors mention the duplication event of NOTCH genes and that NOTCH2 was formed after the split with chimpanzee, whereas Fiddes et al 2018 showed that NOTCH duplication occurred after the split of the orangutans and the other ape lineages.

# It is true that NOTCH duplication occurred after the split of the orangutans and the other ape lineages, but a copy of human NOTCH2 genes acquired a new function by a 4 bp deletion that introduced a fragment of completely new amino acid sequences after the split with chimpanzee.

++ It is very helpful that the text has been updated to avoid confusion.

+ Page 2, second line: “presumably” should be “is speculated to have”.

# Done.

++ Thank you

+ For instance, on page 2, last. Line of the third paragraph, the. authors reference supplemental data, but do not specify which specific supplemental data. When referencing figures or tables, please be specific to make it easier for the reader to follow.

# We have followed the suggestion and revised the MS accordingly.

++ Thank you

+ Figure 2 – how were bin-sizes determined?

# The bin size is 0.1. We mentioned in the legend.

++ Thank you

+ Headers for Figure 2E and 2F are identical.

# The header for figure 2E has been correct.

++ Thank you

+ Figures 2E and 2F should be showing relative number per species rather than absolute. It would also be worthwhile if the authors could speculate on why the distribution of relative Ka/Ks numbers differ between species. Does this relate to their respective mutation rates?

# We do not think they are significantly different, rather they are quite similar to each other.

++ If this is indeed the case, this reviewer suggests to change the figure to clearly reflect that finding. Stacked bargraphs do not allow for easy comparison. Side-by-side bargraphs or better a line-graph showing relative number/species (not absolute, as this is biased based on how many genes were found per species). The last concern is not addressed.

Author Response

see our replies
